# The Resilience of Smallholder Livestock Farmers in Sub-Saharan Africa and the Risks Imbedded in Rural Livestock Systems

Leon Gwaka[ID] and Job Dubihlela *

Faculty of Business & Management Science, Cape Peninsula University of Technology,
Cape Town 7441, South Africa; gwaka@law.upenn.edu
* Correspondence: dubihlelaj@cput.ac.za; Tel.: +27-83-985-5136

**Abstract:** Smallholder livestock farmers (SLF) are important in research and policy development agenda because of the everlasting issue of food insecurity and the livelihoods in sub-Sahara Africa. Lack of access to technologies and information, poor infrastructure and lack of access to markets and environmental factors play a key role in deterring sustainable smallholder livestock farming. In recent years, studies have provided evidence that livestock species can play a role towards solving household food insecurity and enhancing Africa's resilient livelihoods, particularly in rural settings. However, no studies have been conducted on the effectiveness of emerging technologies (available information technologies) as risk mitigation tools for smallholder livestock farmers. The study used survey data from 278 interviewer-administered questionnaires and 13 in-depth interviews village herds from Dumba, Mapayi, Old Nuli and Shabwe to explore whether rural SLF would use the emerging technologies to enhance their operations, and mitigate supply chain risk, exposures to stock theft and loss. Empirical results show the overwhelming need for the use of emerging technologies by the SLF, and that emerging technologies have significant and positive effects on the risk management activities. SLF indicated that, using digital technologies, they could enhance their risk mitigation and value chains. The results of the study have several policy implications. For instance, the agricultural comparative advantage should be improved through various emerging agricultural technologies. Moreover, the immediate rural development service networks for SLF could be strengthened through the Department of Rural Development and Land Reform to reduce livestock risk exposures, disasters and market reactions. Although rural livestock farming in Beitbridge has survived over the years at a subsistence level, the paper provides some interesting and pertinent findings, and projects some possible future research pathways.

**Keywords:** smallholder farmers; livestock farming; resilient livelihood; sub-Saharan Africa; emerging technologies; food security

## 1. Introduction

In the past decade, studies on food security in rural areas of sub-Saharan Africa have increased in response to the intensifying food insecurity challenge which stems from many factors including natural hazards, conflicts, economic crises, diseases and politics [1]. This increase in research on food security has also been necessitated by a need to "substantially improve the evidence base to better inform the design, targeting and implementation of interventions" [2]. The lack of access to technologies and information, poor infrastructure and lack of access to markets and environmental factors play a key role in deterring sustainable smallholder livestock farming [3]. Research has provided evidence that livestock species can play a role in solving household food insecurity and enhancing Africa's

resilient livelihoods [4], particularly in rural settings. However, few studies have been conducted on the effectiveness of emerging technologies as risk mitigation tools for smallholder livestock farmers. Despite the growing body of research, the number of food-insecure individuals and households in sub-Saharan Africa has been increasing, and in recent years studies focusing on finding solutions to the detrimental problem of food insecurity are increasingly arguing that livestock species can play a key role towards solving household and global food insecurity [5]. These claims are, however, met by the stark reality that livestock systems in most sub-Saharan Africa's rural communities, such as Beitbridge in Zimbabwe, are fragile [6]. The fragility of the livestock systems—as well as other key systems—emanates from many factors including the lack of supporting infrastructure, climate change and poor markets [7].

Furthermore, the observed evidence on the fragility of the livestock systems in most rural areas in Africa is overwhelming. For instance, on 29 January 2016, the headline story of the Zimbabwe national newspaper, *The Herald*, read, "16 000 cattle succumb to drought" followed by a second report within the same quarter indicating that "drought claims 24 800 cattle" [2]. Apart from the environment-related challenges such as drought, deaths and diseases, the livestock systems in rural areas also face other challenges such as the lack of competitive markets, poor (sometime, non-existent) policies and lack of actors (buyers, livestock technicians, and advisory services) [8]. Describing the complexities within the African livestock production systems, Lybbert et al. [9] suggest that, " . . . *Livestock production in Africa is characterised by low rates of marketed offtake, aperiodic system crashes in which half or more of the aggregate herd commonly perishes, and complex systems of inter-household livestock gifts and loans to help rebuild herds decimated by climatic or epidemiological shocks*".

Considering the extent to which the challenges facing smallholder farmers in rural communities are intensifying, many suggestions have emerged that there is need for robust risk management strategies to protect household assets and ensure a stable well-being for the rural communities [10]. The subject of risk management in agricultural communities has largely been studied from the perspective of insurance [11], e.g., insurance options and attitudes of smallholder farmers towards insurance. A few other studies agree that, over the years, rural smallholder communities have faced many adversities such as recurrent droughts, resulting in asset losses (e.g., livestock deaths) prompting these communities to develop unique and non-conventional risk management strategies to cope with the challenges they face [6]. However, most recent studies, such as Enenkel et al. [12], concur that, while some of these non-conventional risk management strategies remain in place, the complexities of the risks facing societies (e.g., the unpredictability of system dynamics) have evolved to the extent that new, context-appropriate and advanced risk management strategies are required.

Incidentally, parallel to the intensification of risks facing livestock farmers in rural communities, as well as the need for robust risk management strategies, has been the ubiquity of technological innovations in modern-day societies. These technological innovations, including agriculture-oriented innovations [12], are also penetrating rural communities, even though the majority of these do so through mobile phones [8]. Inevitably, exploring the potential of technological innovations to contribute towards solving societal challenges has become a topical research area [13].

Conversely, the technological and technology infrastructure deficits within the rural communities mean that most technological innovations have been integrated in major sectors such as health and education with limited integration in other local value chains such as livestock systems [11]. Subsequently, there are limited studies available exploring the intersection of technological innovations and local value chain features in rural areas such as Beitbridge. However, with an increase in efforts to address technological deficits in rural communities [12], it is an important challenge to understand how the features of the local value chains—in the context of this study, the risk management processes—in rural communities may be impacted by the development of digital infrastructure in these rural communities and the subsequent integration of technological innovations.

Therefore, this paper attempts to answer the question: can the penetration of digital technologies in rural communities play a role among the livestock smallholder farmers in reducing their risk

exposure? To answer this, the study first explores, using a socio-ecological systems framework, the characteristics of the livestock system in rural communities to identify the potential risks within the livestock system in the manner of Smith et al. [5]. The rest of the paper is structured as follows, following this introduction; the theoretical frameworks guiding the study are discussed before the characterisation of the Beitbridge livestock system using the socio-ecological system (SES) as a guiding framework. The research methodology follows this section and then results, discussion and then conclusions are presented.

## 2. Theoretical Framework

This study explores the interaction of complex concepts relating to risks within livestock systems, for example, the ecology risks, risk management and resilience of smallholder farmers. Exploring these concepts using a single theory can be difficult, thus the study is framed around three theories, which are the socio-ecological system (SES) framework [11], the resilience theory [14] and technology affordances [15]. Essentially, the SES framework suggests that a socio-ecological system, such as a livestock system is made up of attributes—primary and secondary. These attributes interact to produce specific outcomes. In the context of Davis and Chouinard [15], it is purported that the attributes of livestock system and their interactions (or not) are key determinants of the risks faced by smallholder farmers and the subsequent risk management processes.

Therefore, the SES framework is applied to explore the attributes of the study areas' livestock system. In addition, the study applies the resilience theory to explore the " . . . positive contextual, social and individual variables" which " . . . operate in opposition to risk factors, and [individuals] overcome negative effects of risk exposure" [16]. Thus, the resilience theory helps to identify internal (assets) and external (resources) critical for risk management (ibid). Ledesma [14] thinks that applying the resilience theory helps to understand "individual variations in response to risk". Therefore, in the context of the study, the resilience theory helps to understand how certain livestock system actors who were identified using the SES framework respond to risks and succumb to stress and adversity or survive and respond well. Finally, to frame the potential of technological innovations in the risk management processes, the study draws upon the technology affordances theory [15].

### 2.1. Livestock, Risks and Technology in Rural Communities

To develop an answer to the research question set in the introduction, it is critical to develop a picture of the current state of knowledge on the subject(s) being studied and their relationships [11]. Essentially, this section seeks to explore how existing studies frame the livestock evolution [17,18], emerging risks and the potential role of technology towards transforming risk management among smallholder farmers. It is also critical to outline that the study does not assume technology to be passive, as it is shaped by society. Technology innovations could assist in the detection and tests of animal diseases such as foot-and-mouth and avian influenza [15]. Animal pests and diseases are a major threat to livestock and poultry industries and often outbreaks affect access to export markets and undermine livelihoods. There is an ongoing need to practice technology-driven biosecurity. This will help in protecting livelihoods, the environment, and the community from the negative impacts of pests, disease, weeds, and contaminants [13].

### 2.2. Livestock Evolution—A Focus on Risks

Many studies exist on livestock, for instance, there are those that focus on different dimensions such as production, marketing, consumption and history. From these studies, it is evident that, for African families, livestock plays many critical roles. In recent years, there has been a growth in studies exploring livestock's potential to contribute towards household food security [9,19]. However, among these studies, there is also clear evidence on the several differences between livestock systems in developed countries and those in developing countries. Livestock systems in developing countries are characterised by several weak attributes, making their systems susceptible

to many challenges [7]. Climate change remains prevalent in the delineated region, presenting a range of challenges for smallholder livestock farmers (SLF) in Sub-Saharan Africa. Various changes in temperature, the mountainous and arid landscape and water availability impact on the pastures and forage quantities and quality [20], feed-grain pricing, and disease and pest distributions in rural settings.

### 2.3. Technology in Rural Communities—Its Role towards Risk Mitigation

Studies reviewed suggested that the livestock risks in developing countries, rural communities specifically, are concerning. However, cognizant of these risks, Delgado et al. [17] suggest that "technological progress in the production, processing, and distribution of livestock products will be central to the positive outcome of the Livestock Revolution".

In sum, the characteristics of the livestock system in Beitbridge indicate high vulnerability which could explain why, despite being labelled as one of the top livestock-producing districts, households in this region remain poor. The specific characteristics discussed above pose many risks to the livestock farmers in several ways. For instance, the unpredictability of the livestock system means that adversities are likely to occur when farmers are less prepared [19], which can result in huge losses —such as losses to the *El Nino* phenomenon. The ultimate dependence on livestock systems by most of the households also means that the poor performance of the livestock system affects many households [6], impoverishing the greater parts of the community [10]. Therefore, since the actors, namely smallholder farmers, are poor and depend on the livestock system extensively, and yet the system is vulnerable, it is critical to explore the risks among livestock-dependent households. In the next section, the research methodology on what was done as part of efforts to answer the questions is presented.

## 3. Research Methodology

### 3.1. Research Design and Study Area

This study was conducted in Ward 15 of the Beitbridge Rural District, Zimbabwe. The study area comprises four villages, namely, Dumba, Mapayi, Old Nuli and Shabwe. Statistical data from the study by Das Nair, Chisoro and Ziba [18] show that the four villages have 932 households, with a total population of 4166, resulting in an average household size of 4.5 people per household. Of the total enumerated people in the Ward, about 54% are female and 46% are male [4]. This study adopted a mixed method case study approach to generate data from four rural villages in Beitbridge District, Zimbabwe. Thus, data used for this study were gathered using qualitative and quantitative research methodologies over a period of 3 years.

### 3.2. Sample Size

The selection of participants for qualitative data in this study was systematically planned. For instance, the study started by observing the whole community engaging in livestock auction, then narrowed down to community visioning workshops held every three months in 2019, whose aim was to know the key participants for surveys and inform communities about the project through the village herdsmen. The participants engaged in the community visioning workshops were drawn from the four villages, specifically the village herds. Further, from those who participated in the community visioning workshops, participants for the focus group discussions and key informants for in-depth interviews were identified. The intensity of engagement increased as the in-depth interviews held in September of the same year drew more interest from participants.

To obtain quantitative data, a survey-questionnaire was administered between September and December 2019 to a sample of 278 households. Further to this, qualitative data were obtained through ethnographic research methods including focus group discussions and in-depth interviews. The study utilised participatory risk mapping [21]. The community described their lived environment and the livestock system using words (discussions), text (individual and group notes), diagrams and maps.

Research participants identified several risks relating to livestock and others impacting the general livelihood. These were then collated, and each risk was rated and ranked by the community. The process of risk ranking involved two charts depicting likelihood and the impact of risk. On each chart, it was split into two—(high risk, low risk) and (high likelihood and low likelihood). Stickers (colours red and green) were also used in the process. For each identified risk (e.g., lack of water source, with the risk being human and livestock starvation), it was numbered. When a risk was read by the facilitator, each participant selected a stick-on note (red, orange and green) and pasted it on one column. Red symbolised high likelihood, orange represented medium impact, and green low likelihood (on the likelihood chart), then red for high impact and green for low impact on the impact chart.

## 4. Results and Discussion

### 4.1. Livestock as a Livelihood Option

Most rural areas in developing countries practice diversified livelihood options. The practice of diverse livelihoods is even encouraged within existing studies, as a strategy to overcome the fragility of some livelihood options. For instance, studies have suggested that in rural communities, there is need to move towards more non-farm enterprises since, in the past decade [19], climate change has adversely affected the farm output, subsequently affecting the livelihood of those solely depending on farm output [18]. The dependence on a specific livelihood is determined by several factors including the gender of the household head, education and age [4]. In this study, when respondents were asked about the household livelihood options, results, as presented in Table 1, it was revealed that livestock farming is a core livelihood option for the people living in rural Beitbridge. This farming method has been passed from generation to generation, albeit without much enhancement in terms of operation, risk mitigation or value chain adaptations [22].

**Table 1.** 2019 Livelihood options in Beitbridge and corresponding seasons.

| | January | February | March | April | May | June | July | August | September | October | November | December |
|---|---|---|---|---|---|---|---|---|---|---|---|---|
| Livestock auctioning | 251 | 251 | 251 | 243 | 231 | 201 | 198 | 253 | 246 | 263 | 186 | 73 |
| Mopane worms | | 183 | 197 | 186 | | | | | | | 79 | 47 |
| Baobab selling | | | | | 103 | 119 | 97 | 143 | 159 | | | |
| Gardening | | | | 158 | 113 | 79 | | | | | | |
| Border related activities | 87 | 91 | 79 | 137 | 121 | 106 | 86 | 93 | 101 | 97 | 126 | 207 |
| Farming | | 213 | 213 | | | | | | | 213 | 213 | 213 |
| Remittances | 96 | 96 | 96 | 103 | 103 | 103 | 97 | 97 | 107 | 135 | 176 | 219 |

Despite cross-border related activities and remittances being other sources of income throughout the year [4], many households, regardless of gender or household head, indicated that livestock was central to household income generation and household food security. The importance of the livestock from the study findings was also reiterated in several other studies and, most notably, the HLPE (2016) and Mottet et al. [22]. During the 2016 CFS convention, livestock was ratified as critical to household food security across Africa. One major distinction of livestock farming to livelihoods in the Beitbridge District area, like other societies, is the SLF's ability to contribute to multiple livelihoods dimensions of economic, social and religious importance [19].

Since different social groups pursue different income-generating activities, the study sought to segregate the economic activities by age. The indicators from the Pearson chi-square test reported in Table 2, show significant differences ($p = 0.009$, $p < 0.05$) between individual age groups and their active economic pursuance. Most individuals in the age groups 18–25 (6.3%) and 26–35 (8.3%) were less likely to engage in income generation activities related to natural resources, while respondents in the age group above 46 years were more inclined towards the natural-resources-related income generation activities, which include the harvesting and selling of baobab fruits, selling mopane worms, weaving grass-mats and customary woodworks.

**Table 2.** Demographic profile of Beitbridge smallholder livestock farmers.

| | | | Natural Resources | Petty Trading | Community Gardens | Total |
|---|---|---|---|---|---|---|
| Age in years | 18–25 | N | 2 | 15 | 15 | 32 |
| | | % | 6.3% | 46.9% | 46.9% | 100.0% |
| | 26–35 | N | 4 | 29 | 15 | 48 |
| | | % | 8.3% | 60.4% | 31.3% | 100.0% |
| | 36–45 | N | 7 | 19 | 16 | 42 |
| | | % | 16.7% | 45.2% | 38.1% | 100.0% |
| | 46+ | N | 25 | 29 | 30 | 84 |
| | | % | 29.8% | 34.5% | 35.7% | 100.0% |
| Total | | N | 38 | 92 | 76 | 206 |
| | | % | 18.4% | 44.7% | 36.9% | 100.0% |

N = number of farmer respondents per catergory.

Apart from the age differentials, the results in Table 2 also showed that women were considerably more involved in livestock farming systems, particularly during the livestock auctions. This came as a surprise because, in emerging markets, livestock farming is historically and predominantly viewed as a preserve for their male counterparts [23]. During the participant observation sessions, more women were observed selling livestock or contributing to the pricing of the livestock. When one participant was asked for the potential reason for the considerable inclusion of women, as many studies show that women are generally excluded, the respondent indicated that, " … *In this Beitbridge District area, many men migrate to neighbouring countries, particularly South Africa in search of better opportunities. As a result, women take the responsibility and get involved in livestock production and marketing. However, most women still act based on instructions sent by their husbands that explains why it is difficult for the buyers to negotiate prices with these women than their male counterparts* … ".

*4.2. Characterising the Beitbridge Livestock Systems*

Characterising intricate systems such as rural livestock systems is complex but helps to understand the dynamics within the systems. If risks within the livestock systems are to be well understood, it becomes imperative to closely examine the livestock systems. Davis and Chouinard [15] suggested that, often, complex systems are misunderstood due to a lack of common frameworks that dissect the intricacies within these complex systems. Subsequently, the social, economic and political (SES) framework was developed as a tool to explore the dynamics of such a complex system. The SES framework has continuously been improved and applied to various systems, including fisheries, forests, and water resources [24]. Further to this, the SES has even been applied to human-constructed systems such as telecommunication systems and power grids. It is Marshall's [24] work on a livestock system in Cambodia which underlines the applicability of the SES to the analysis of a livestock system.

The SES framework proposed by Marshall [24] suggested four key pillars of livestock systems as illustrated in Figure 1. Each pillar has secondary attributes which help to explore the complex livestock systems. The Actors [A] pillar, whose secondary attributes include the number of actors, socio-economic conditions of actors, and dependence on SLF. From the extensive ethnographic methods done during field work, it emerged that the socio-economic attributes of SLFs and those of the buyers are incongruent. The actors also do not make use of technologies [13].

The resource units (RU) pillar revealed the significant economic value of livestock for rural households, which is why some hold more than 50 cattle each. The large numbers of livestock, while representing higher economic value for such SLF, impacts negatively on grasslands rehabilitation and sustainable livestock grazing. It also has an impact on rural eco-system and biodiversity. From a governance systems (GS) perspective, there are rules-in-use specifically relating to livestock theft,

mitigating the exposures of the SLF in the manner of Andaleeb, Khan and Shah [23]. Policies are unclear at different levels for the control of the livestock systems, from the Department of Rural Development and Land Reform (DRDLR) down to village committees. Most parts of the livestock system still lack the simple implementation of rules governing the smallholder livestock farming and their markets.

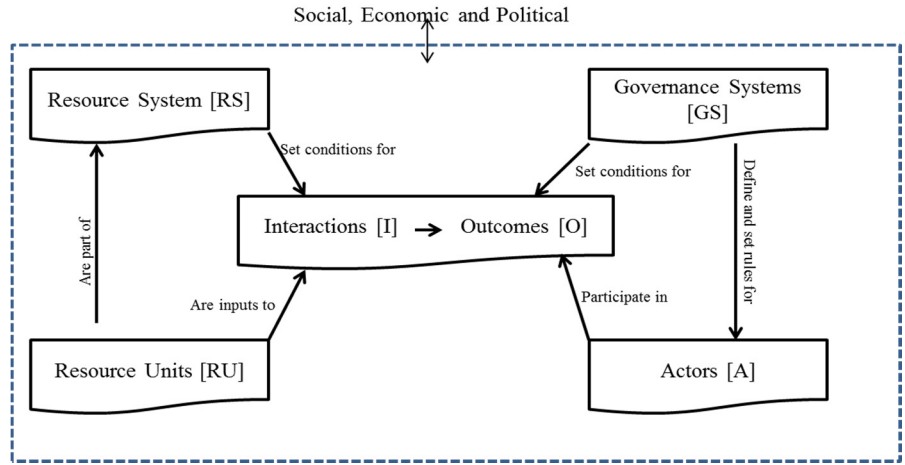

**Figure 1.** The SES framework [24].

### 4.3. System Attributes as Risk Factors

System attributes of the SLFs showed a dire lack of human constructed livestock infrastructure, limited actors (buyers), and limited to non-use of emerging technologies. These findings clearly suggest that SLFs in Beitbridge District area are exposed to risk exposures that they cannot abate. Their lack of risk definition is something that increases their susceptibility to all forms of risk [3], 2017). During the surveys, the interviewer engaged the respondents in risk identification and risk ranking. Consistent with Bailey et al. [21], this study found that the research participants are exposed to risk of theft, operational risk, sub-optimal livestock market-value and high livestock mortality.

### 4.4. Risk Mitigation and the Smallholder Livestock Value Chain

Empirical evidence presented in Section 4.3, as reported in Table 3, show the different risks which SLFs in Beitbridge District encounter. However, according to Delgado et al. [17], despite the several risks faced by the SLF, such farmers have encultured unique mechanisms to identify, and attempt to mitigate ongoing risk exposures. Mottet et al. [22] asserts that risk assessment is characterised by identifying and ranking the different risks which, in the context of this study, are risks in the livestock system owned by the communal SLFs. Many communities have adopted personalised involvement (roles) of the community members as the yardstick in the risk assessment [23]. While the involvement of communities is particularly imperative for SLF in Beitbridge District, it has, however, revealed the need for affected communities to have access to emerging technologies. They are receptive to these emerging technologies and motivated to make the necessary adjustments in their operations.

The empirical evidence show that in terms of risk mapping, livestock smallholder farmers have extremely limited capacity to identify new or emerging risks and in recent years, many new diseases (risks) have been emerging [25]. In such cases, the agricultural extension workers lag in communicating disease risk mapping due to limited technological adoption patterns. The results also revealed the limited knowledge which farmers have on livestock growth patterns, livestock diseases, livestock commercial markets [25]. The respondents indicated that the little that they know is mostly gained from peers and training by the local extension workers (very infrequent due to lack of resources). In both cases, the upgrading of knowledge is infrequent, thus, risk mapping is at extremely low levels. The results further support Plough and Krimsky [26], who argued that risk communication must inform individuals about the existence, nature, form, severity, or acceptability of risk exposures.

In this light, a migration from hard manual operations and poor communication to emerging digitised technologies would greatly enhance the risk mitigation activities of SLFs in rural settings.

**Table 3.** Risk factors and likelihood impact on Smallholder Livestock Farmers.

| Risk Factor (Attribute) | Risk | Impact | Likelihood |
|---|---|---|---|
| Few livestock actors (lack of buyers) | Market/Price risk: *lack of buyers* | High Impact: Farmers receive substantially lower (sub-standard) prices for their livestock | High Likelihood: In the past 5 years, farmers have always been receiving very low prices |
| System dynamics unpredictability | Production Risk: *poor quality* | High Impact [8/10]: Farmers receive substantially lower (sub-standard) prices for their livestock | High Likelihood [8/10]: In the past 5 years, farmers have always been receiving very low prices |
| | Mortality Risk: *livestock deaths* | High Impact: Livestock death is a total loss | Medium Likelihood: Livestock die but also, livestock reproduce |
| Poor policies | Theft—direct loss | High Impact: Livestock theft is a total loss | Medium Likelihood: Livestock die but also, livestock reproduce |
| | Effect of Gender: *Female/male owner* | High Impact: Farmers receive substantially lower (sub- standard) prices for poor quality livestock | High Impact: Farmers receive substantially lower (sub- standard) prices for poor quality livestock |
| Limited/Poor Human constructed facilities | Production Risk: | High Impact: Livestock death is a total loss | Medium Likelihood: Livestock die but also, livestock reproduce |
| | Market Risk | High Impact: Farmers receive substantially lower (sub-standard) prices for poor quality livestock | High Likelihood: In the past 5 years, farmers have always been receiving very low prices |

### 4.5. Risk Sharing and Risk Transfer Options

SLFs have options which they can pursue to transfer risk to a third party normally through insurance, but the rural setting has prevented rural livestock insurance from being successful in Southern Africa [27]. In the context of this study, risk transfer may be more applicable to operational risks. Exploring risk transfer for farmers, Chantarat et al. [27] posited that insurance has mainly stayed in environmentally risky areas. In this case, when the study participants were asked about insurance of their livestock and household assets, 94% of the respondents reported having no insurance at all. One of the respondents as quoted stated that, " . . . *This thing of insurance is very expensive—you must pay every month and yet, often times we do not have stable and predictable monthly income. Defaulting on premiums often results in the loss of all premiums previously paid. There is no comprehensive cover for us, and the insurance companies require that we always have airtime when we make claims* . . . ".

It is not surprising that Mmakgabo [3] suggested that smallholder farmers in rural settings are willing to take insurance if available and designed appropriately to meet their needs. For this study, due to the massive deaths of livestock in recent years, and the prevalence of theft incidences, SLFs showed a willingness to take up risk-abating emerging technologies, and to share indigenous knowledge and risk transfer information. Such risk control options, if driven by emerging technologies, become essential tools for SLFs in the age of environmental challenges, diminishing grasslands and increasing crime levels.

### 4.6. Emerging Technologies and Market Risks

In relation to sub-optimal prices, Fernandes et al. [11] suggested that the implementation of the livestock traceability system would enable farmers to explore commercial and export markets, although characterised by complexities. Apart from playing a critical role in risk-sharing, technological advances can also aid risk-avoidance, particularly for the dynamic market risks [28]. Chantarat et al. [27] suggested that producer price risk could be handled through policy or project interventions aimed at

bringing efficiencies in price communication, such as market information portals and price broadcasting services. Interestingly, this study confirms for the Beitbridge District that SLFs vastly remain without access to emerging digital technologies, in spite of the growing penetration of digital technologies in Southern Africa. The introduction of emerging digital technologies, mobile phones in particular, could enduringly assist SLFs to "overcome remoteness" and mitigate market risk.

## 5. Main Implications

For instance, the agricultural comparative advantage should be improved through various emerging agricultural technologies. Moreover, the immediate rural development service networks for SLF could be strengthened through the Department of Rural Development and Land Reform to reduce livestock risk exposures, disasters and market reactions. The study has demonstrated the interlacing of risks in Table 1. Based on this, the study suggests that SLFs must have a broader perspective of risk and adopt a system-wide risk mitigation view in their operations. This can be achieved through concerted training and mentorship programs. There is a need for interventions that will enable SLFs to understand that their risk exposures do not relate to livestock production activities alone [11]. A comprehensive view of their value chain may result in improved risk mitigation activities across rural communities, with the potential to decrease the vulnerability of SLFs to market-related risks by using emerging technologies. To this end, communities are encouraged, although they have limited resources, to adopt risk mitigation systems such as the traceability system which helps to counter both production and market risks. Thus, technology-enabled system-wide risk management approaches have the ability to deliver improved risk control for SLFs [28]. Both the DRDLR through the Livestock Production Department (DLPD), and civic society could play a big role in a multi-faced approach of resourcing the technology infrastructure, regulating the sector-market and skills development.

In addition, in rural farming communities of Beitbridge District, market and price risks may continue to persist if there are no policy guidelines to support SLFs. The current system in livestock auction markets in rural communities is the 'spot' market and alternative market arrangements (AMA). The AMA refers to all possible alternatives to the cash market, including arrangements such as forward contracts, customer agreements and packer ownership [29]. The AMA exposes the SLFs to extreme market and price risks which they cannot control.

In this context, technological innovations may be used to bring efficiencies, although training and education of the rural SLFs who often have natural resistance to change will be imperative [6]. The study also confirmed that SLFs do not have an appreciation of anticipatory risk mitigation techniques. Given that SLFs are already resource-constrained, their reactive approach is detrimental to the wellbeing of their operations and the food security and livelihoods of rural households. Considering that, by its nature, livestock replacement rate is slow, anticipatory risk prevention tools that could deter risk exposures would be recommended [11]. The study also recommends the use of positive deviance approaches that look at those SLFs that are doing well and enhancing their resilient indigenous know-how to achieve better results under constraining conditions. This can be enabled by coordinated approaches that tap into their already-existing leadership structures, risk-sharing approaches and communal knowledge bases.

## 6. Conclusions

Although rural livestock farming has survived over the years at a subsistence level, the paper provides some interesting and pertinent findings, and projects some possible future research pathways. The empirical results show the overwhelming need for the use of emerging technologies by the SLF, that emerging technologies have significant and positive effects on the risk management activities. SLF indicated that using digital technologies, they could enhance their risk mitigation and value chains. It is also this study's submission that reactive risk-mitigation techniques (post-risk occurrence) is the dominant risk management approach. The findings of this study are critical to livestock system actors. During the study, interviewers were occasionally accompanied by the relevant departmental

staff from the Livestock Production Department (DLPD). The technicians acknowledged the value of gathering information about the risks that SLFs are exposed to and the need for market developments within the existing livestock systems. This study has its own shortcomings. It is known within the risk management domain that predictive models as well as longitudinal designs are common; this study opted to use a cross-sectional mixed research method. The question of whether adequate and quality data for modelling could be easily obtained played a major role in the decision. Further research could adopt robust predictive models that can add to this study. It is hoped that, with further research in other regions, more data could be available in order to apply alternative statistical tools.

**Author Contributions:** The two authors equally contributed to this research article. L.G. did the first draft and data collection. J.D. did the methodology and the reviews as well as the final editions of the manuscript. All authors have read and agreed to the published version of the manuscript.

**Acknowledgments:** Authors would want to acknowledge reviwers' efforts and guideline in polishing this article, and the employer (Cape Peninsula University) for the time spent and opportunity to write and publish this research project.

**Conflicts of Interest:** The authors would like to declare that there are no potential conflict of interest, nor are there any personal circumstances or interests that may be perceived as inappropriately influencing the representation or interpretation of reported research results.

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
