# Peer review of "The Resilience of Smallholder Livestock Farmers in Sub-Saharan Africa and the Risks Imbedded in Rural Livestock Systems"

_agriculture, doi:10.3390/agriculture10070270_

Round 1

Reviewer 1 Report

Authors: This is a very important study in that it addresses the importance of dealing with changing climates and livestock production impacts. Most especially it starts the conversation about the missing links in the information chain that are needed to ensure future survival of the populations involved, v/v the use of livestock as both economic value and food security.

Many issues related to the ineffectiveness of the extension service due to resource constraints are raised and this work should help put that critical need into recognition. 

I recommend that greater emphasis on the importance of changing climate on incidence of drought and changes in disease and insect prevalence be made. That will strengthen the impact of the work.

Editorially, there are many references cited in the text that do not appear in the list of references, and that needs to be corrected. 

Basically a very sound piece of work and it needs to be made available. Thank you for your efforts!

Author Response

  • Authors are greatly appreciative of the comments and suggestions of the reviewers and have revised the manuscript in line with the reviewers' comments

  • Although the study was not particularly focused on climate and diseases, the recommendations of the reviewers have been adhered to. The revised version has incorporated the emphasis on impact of climate change as well as the impact of diseases and pests in Sub-Saharan Africa smallholder livestock farming.

  • Editorially: Paper has been revised to ensure that all the in-text references are appearing in the reference list as per the reviewer's guidance

  • Paper has also been revised to ensure that academic language is improved very much.

Reviewer 2 Report

Introduction

Your introduction should be logically explain why your research question is important. You have added too much of information to your introduction. Thus the reader can not follow what you exactly want to highlighted. Better to rewrite the introduction following in logical sequence

Line 60 - 70 - Research question
can the penetration of digital technologies in rural communities impact the risk management strategies among the livestock smallholder farmers?

Your research question is too broad. You have not explain well what were your specific objectives of the study. to answer the research question.

Line 75- SES- you should introduce what is SES here. Otherwise reader will confuse

Line 84-85- need reference to prove the interactions.

Method

lines 159 - 162-

Need to explain why color needed to find answers?.Did the participant was alone when responding to the question? or as a group? were there any byers to the respondent?

Results and discussion

Line 175- what do you mean by "this farming method"?

Table caption-  table captions are not well describe the tables. You should improve them.

Table 1- Do these data represent all 3 years. If so you need to say that in the caption

Line 189- Chi-square test is not applicable here. You should explain your statistical methods in the methodology section.

Table 2- you have not included results for gender here. But you are talking about gender effect. You should include gender effect also in the table 22.

Your discussion need more improvement following scientific writing.

Conclusion

Conclusion is not accordance with the methodology and results.

Author Response

  • Authors greatly appreciate the guidelines and comments of the reviewers, and have revised the manuscript in line with the recommendations:

  1. Sections in line 60-70 have been revised to indicate that the digital innovations could be employed within the rural setting to enable them track and reduce exposures. Reviewer had asked if digital penetration can influence risk management strategies in rural settings. 
  2. The research question has been narrowed to focus on the main objective of the study, that is the role that digital technology can play in reducing small holder farmers exposure within rural settings
  3. Line 75, the abbreviation was rectified in line with reviewers comments, SES framework first written in full in the revised version
  4. Line 84-85, the reference has been incorporated for ease of interactions and readership
  5. Line 159-162, methods: the methodology has been revisited and revised in line with reviewers. the colour coding was merely important for classification of likelihood of occurrences and to make it easy for the respondents. it was also essential as a codification for the analysis that followed the data collection.
  6. Line 175 has been revised to read as 'The peasantry livestock farming has been passed from generation to generation....'
  7. Line 189, Chi-square has been removed and the section revised in line with comments from reviewer
  8. Comments of the reviewer well-taken and incorporate in the overall write-up of the manuscript

Round 2

Reviewer 2 Report

Line 157- Expand SLF. This is the first time that you are using this.

Line 251 and line 302- Expand SLF in the title of the table 2 and Table 3

Line 230- Mention the data collection time period (year/ number of years) for the table 1

Authors should explain why they did change citations from the previous citations- Line 128,191,267,340,348,352

Author Response

We greatly appreciate the comments of the reviewers as they assisted in the refinement of the paper.

  1. Line 157, expand the abbr SLF; There is no need to expand this one as it is already expanded in line 145, page 4, para 1 second sentence
  2. line 251 and 302, expand SLF in the table; changes have been effected
  3. line 230, data collection timing; changes have been effected to reflect timing of data collection 
  4. line 128, 191, etc; changes in citation was done to ensure use of more recent sources, also there were major changes in the last round; was an opportunity to keep the flow of the article and reference list as crisp as possible.